# CFBD: Coarse-to-Fine Detection of Backdoor Attacks in Multimodal Contrastive Learning

## Abstract

The backdoor attack in Multimodal Contrastive Learning (MCL) task has been receiving increasing attention in recent years, due to numerous downstream tasks that rely on pre-trained MCL models. Backdoor detection has been one of the effective protection solutions to fight against backdoor attacks. However, the majority of existing backdoor detection methods in MCL usually produce non-satisfying detection results. Two main factors are responsible for this: 1) one-stage detection lacks subsequent dynamic adaptation to the distribution of poisoned and benign pairs when faced with different attacks, and 2) the criteria used in existing methods, specifically the cosine similarity between image and caption, is insufficient to distinguish between poisoned and benign pairs. To address these problems, we extend the conventional one-stage detection architecture to a two-stage architecture and propose a better metric in the second stage with high precision and high fault tolerance. To this end, we design a novel **C**oarse-to-**F**ine two-stage **B**ackdoor **D**etection method, termed CFBD, which primarily focuses on multimodal learning involving image-caption dataset, such as CLIP. The objective of the coarse-grained stage is to roughly partition the dataset into poisoned, benign, and suspicious subsets. In the fine-grained stage, we use the average textual correlation with the poisoned subset to improve the detection quality. Extensive experiments demonstrate that CFBD achieves superior backdoor detection performance, e.g., almost 100% True Positive Rate (TPR) for various attacks over the large-scale dataset CC-3M, markedly outperforming state-of-the-art methods.

## 1 Introduction

Multimodal Contrastive Learning (MCL) represents a pivotal advancement in the field of deep learning, specifically within the realm of learning from different modalities. This approach leverages the synergistic integration of different data modalities to learn robust and generalizable representations. Multimodal contrastive methods, e.g., CLIP Radford et al. (2021a), ALIGN Jia et al. (2021) and BASIC Pham et al. (2023), have shown impressive results in various downstream tasks such as text-guided image generation Kim et al. (2022); Ramesh et al. (2022); Clark & Jaini (2023); Nichol et al. (2022); Avrahami et al. (2022); Wang et al. (2023b); Ye et al. (2024) and video understanding Liu et al. (2023d); Zhang et al. (2023); Li et al. (2023b); Maaz et al. (2023). Generally, MCL models use a contrastive optimization objective to attract matched image-caption pair representations closer together in the embedding space while repelling unmatched pairs. Additionally, the advent of MCL models provides an advantage to developers with limited resources, allowing them to build high-quality models for downstream tasks by fine-tuning readily available pre-trained MCL encoders, such as CLIP. Without loss of generality, we focus on the CLIP models in this work, and our developed detection technique can be easily generalized to other MCL models.

MCL harnesses vast internet-sourced image-caption datasets for improved semantic understanding, this reliance also introduces notable vulnerabilities. As elucidated in key studies Carlini et al. (2024); Carlini & Terzis (2022); Yang et al. (2023c), large-scale models that utilize these massive datasets are especially vulnerable to targeted data poisoning and backdoor attacks. Adversaries can execute a backdoor attack by injecting specialized triggers into a small subset of training images and modifying their original captions (*e.g.*, "A deer with a white background") to target captions (*e.g.*, "An image of cat"), as depicted in Figure 1. During the pre-training phase, the contrastive loss objective, designed to align the embeddings of congruent image-caption pairs, inadvertently also aligns

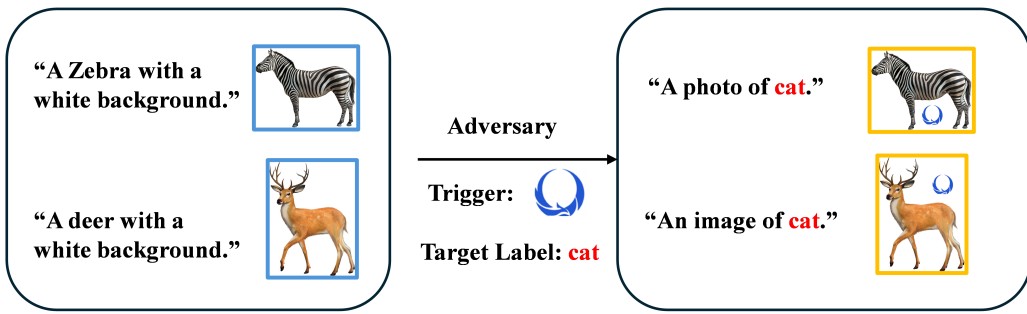

Figure 1: The strategy employed by the adversary to introduce backdoor attacks into the model. It injects a backdoor trigger to clean images and changes their original captions to target captions for the target label (in this case, "cat").

those of the poisoned pairs. This misalignment results in a spurious correlation between the triggered images and the target class (*e.g.*, "cat" in Figure 1), compromising the integrity of the model. Such manipulations highlight the profound security challenges that multimodal models encounter, emphasizing the urgent need for effective defensive solutions.

There are also some attempts to protect multimodal learning against backdoor attacks. A common assumption adopted by these methods is that for poisoned pairs, the dissimilarity between image-caption pairs is larger than that of those benign pairs. Based on this assumption, (Yang et al., 2023c) explored the use of a benign pre-trained CLIP model, to filter out dissimilar image-caption pairs. (Bansal et al., 2023) developed CleanCLIP, employed an in-modality contrastive loss for both visual and textual modalities, aiming to neutralize backdoor influences. (Yang et al., 2023a) observed that the poisoned images and captions are not close to groups of similar images and captions in the representation space, early in training. They intend to replace the original captions with the nearest captions in the dataset based on the image representation. However, the substitution is carried out for both benign and poisoned pairs, even if the ASR can be fairly low, the model performance will deteriorate along with the substitution with benign pairs. (Yang et al., 2023b) applied unimodal contrastive learning separately to each modality, categorizing data into "safe" and "risky" subsets to prevent the injection of backdoors. A recent study (Liang et al., 2024c) also strengthened backdoor shortcuts to identify suspicious samples through training prioritized by weakly similar samples. More introductions on related works can be found in the Appendix B. Most of these methods perform a one-stage detection process to filter out poisoned samples and safeguard the training, in which the detection is designed under the assumption of dissimilarity in poisoned pairs.

Despite their impressive results, there is still room for improvement. In this work, our aim is to design an effective detection method to safeguard the training of CLIP models. We start by pinpointing a prevalent problem in current solutions that one-stage detection is inadequate to accurately discriminate poisoned pairs within the poisoned dataset. Specifically, these methods regularly mis-identify the benign pairs as poisoned pairs. The inaccurate separation will precipitate a high Attack Success Rate (ASR) in the post-training model, whereas the mis-identification problem could risk undermining the inference performance of the model on benign input. On the other hand, utilizing the high cosine dissimilarity between image and caption as a sign of being poisoned struggles to effectively detect poisoned pairs. As can be seen from the distribution of cosine similarity in Figure 2 (b), there is a large overlapping region between the distribution of poisoned pairs and benign pairs, indicating that the cosine similarity between two modalities is insufficient to separate poisoned and benign pairs. We consequently adopt an experimental approach to justify that textual correlation could be a better metric for identifying poisoned pairs. In Figure 2 (a), we give the textual correlation between different poisoned and benign pairs. The captions in poisoned pairs exhibit high similarity in the textual embedding space, significantly overwhelms that of between benign pairs. Assume now that we are given a poisoned set $\mathcal{D}_p$, though this set is not required when launching the detection. We then calculate the average textual correlation with captions in this set for each pair in a poisoned dataset. As can be seen from Figure 2 (c), the textual correlation with respect to a given poisoned set could be a more effective metric for discriminating the poisoned pairs from the benign pairs. To this end, we propose a two-stage detection process in which a coarse-grained stage is employed



Figure 2: (a) textual correlation score involving poisoned and benign pairs. (b) distribution of cosine similarity for benign and poisoned pairs. (c) distribution of textual correlation with poisoned captions for benign and poisoned pairs.

to collect benign subset and poisoned subset for the later fine-grained detection stage. To be more specific, in the coarse-grained stage, we maps the visual embedding to the textual embedding space given images in the dataset, and applying a Gaussian Mixture Model (GMM) to fit the similarity of image with the original caption and synthetic embedding, respectively. In the fine-grained detection phase, the pairs in the suspicious subset are further classified as poisoned or benign, according to this textual correlation. Extensive experiments demonstrate that our CFBD can effectively detect poisoned pairs from the training dataset, and thereby prevent the injection of backdoors in the subsequent training process. It is noted that CFBD achieves a true positive rate (TPR) of nearly 100% and a false positive rate (FPR) of nearly 0% in detecting 3000 poisoned pairs from a large-scale dataset CC3M. Our major contributions can be summarized as follows:

- We extend the mainstream one-stage detection architecture into a coarse-to-fine two-stage detection architecture which yields improved detection results. E.g., for BadNets, Blended, and Trojan attacks on the CC3M dataset, CFBD can reach 100% TPR and 0% FPR.

- We propose a more effective metric in the fine-grained detection stage, outperforming the widely-used image-caption similarity metric. Notably, this metric demonstrates significant fault tolerance with the coarse-grained detection result.

- Extensive experiments demonstrate CFBD achieves superior performance, significantly outperforming state-of-the-art methods, in terms of detecting poisoned pairs and preserving the model accuracy.

## 2 PROPOSED BACKDROOR DETECTION METHOD CFBD

Before diving into the details of CFBD, let us first explain the threat model.

**Threat Model.** We consider a paired image-caption dataset $\mathcal{D} = \{(I_i, T_i)\}_{i=1}^N$, where $I_i$ and $T_i$ denote the image and the associated ground-truth (GT) caption, respectively. The dataset $\mathcal{D}$ can be divided into two subsets: $\mathcal{D}_p = \{I_i, T_i)\}_{i=1}^n$ and $\mathcal{D}_b = \{(I_i, T_i)\}_{i=n+1}^N$, corresponding to the potential poisoned and benign data, respectively, where $n \ll N$. The adversary is allowed to access and manipulate $\mathcal{D}_p$ such that images carrying the trigger $t$ are misclassified into the target classes $y$, while other images are classified correctly. To this end, with $\mathcal{D}_p$, the adversary crafts a poisoning subset $\mathcal{P} = \{(\hat{I}_i, T_i^y)\}_{i=1}^n$ through a generic trigger adding process $\circ$, namely, $\hat{I}_i = I_i \circ t$ and replacing the GT caption $T_i$ with target caption $T_i^y$. Eventually, the CLIP model trained on the combination of poisoned dataset $\mathcal{P}$ and the benign subset $\mathcal{D}_b$ spuriously associates the presence of the trigger $t$ in an image with the target label $y$ in the target caption. It is also assumed that the adversary knows the model structure, the training algorithm, and the employed hyperparameters, but has no control of the downstream training process.

**Detection Goal.** Fundamentally, the objective of detection is to clearly detect poisoned set $\mathcal{D}_p$ from the poisoned dataset $\mathcal{D}$. Ideally, the multimodal model trains on the remaining benign pairs should have a low ASR, while maintaining a high inference performance on the benign test.

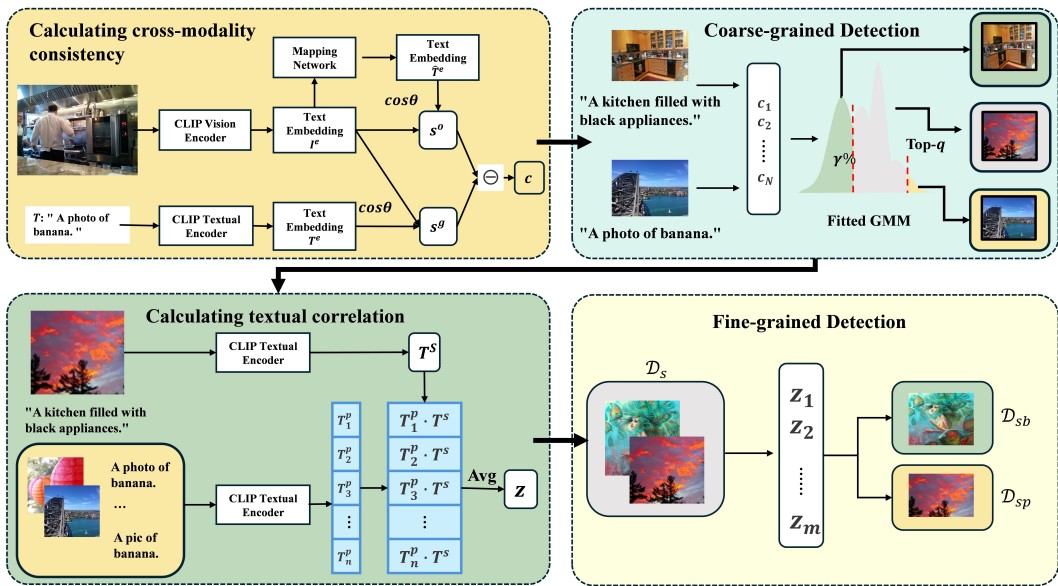

Figure 3: The proposed coarse-to-fine backdoor detection framework.

## 2.1 METHOD OVERVIEW OF CFBD

The schematic diagram of our proposed CFBD is illustrated in Figure 3. Essentially, CFBD is a coarse-to-fine backdoor detection method consisting of two stages: a *coarse-grained detection stage* and a *fine-grained detection stage*. In the coarse-grained detection stage (see Figure 3 (a)), the image $I_i$ is first processed through a pre-trained CLIP vision encoder $f_I$ to obtain the image embedding $I_i^e = f_I(I_i)$. Then $I_i^e$ is sequentially processed by a mapping network $F$ to generate a synthetic textual embedding $\hat{T}_i^e$. We define $s_i^o$ as the cosine similarity between the pair $(I_i^e, T_i)$ as the original similarity, while $s_i^g$ as the cosine similarity for the pair $(I_i, \hat{T}_i^e)$ as the generated similarity. Then the cross-modality consistency $c_i$ can be computed as the difference between $s_i^g$ and $s_i^o$. Upon obtaining consistency values $\{c_i\}_{i=1}^N$ for all the pairs in dataset $\mathcal{D}$, we adopt a GMM to fit all these consistency values. Based on the fitted GMM, we aggregate the pairs with the probability higher than $\gamma\%$ generated from a specific Gaussian distribution into benign subset $\mathcal{D}_b$. The pairs with the top-$q$ $c_i$ values are grouped as poisoned subset $\mathcal{D}_p$, while the remaining pairs (less confidence level) are categorized into the suspicious subset $\mathcal{D}_s$.

The fine-grained detection stage (see Figure 3 (b)) is devoted to further identify poisoned pairs in $\mathcal{D}_s$. Specifically, for an unidentified pair $(I_u, T_u)$ from $\mathcal{D}_s$, we calculate the average textual correlation $z$ between the caption $T_u$ and the collection of captions $\mathcal{T}_p = \{T_j \mid (I_j, T_j) \in \mathcal{D}_p\}$ from the identified poisoned subset $\mathcal{D}_p$. Afterwards, this pair is classified as poisoned pair if $z$ is larger than a threshold $\gamma_z$, motivated by the factor b) in Section 1. Otherwise, it is classified as a benign pair. By the end of the fine-grained stage, the suspicious subset $\mathcal{D}_s$ is split into a poisoned part $\mathcal{D}_{sp}$ and benign part $\mathcal{D}_{sb}$. Eventually, a CLIP model can be trained with $\mathcal{D}_b \cup \mathcal{D}_{sb}$. The full Algorithm is in Appendix A.

We are now ready to give the details on the coarse-grained and fine-grained detection stages.

## 2.2 COARSE-GRAINED DETECTION VIA CROSS-MODALITY CONSISTENCY

**Visual-guided Text Embedding Generation.** Since the visual feature space and textual feature space are normally not aligned, we propose to use a mapping network $F$ to mapping the visual embedding $f_I(I_i)$ to the textual domain. Similar to Li et al. (2022a), we apply a language modeling (LM) loss to maximize the likelihood of the text in an autoregressive manner. This LM loss enables the model with the generalization capability to convert visual information into coherent textual embedding space. Given the image embedding $I_i^e$, the mapping function $F$ projects the visual

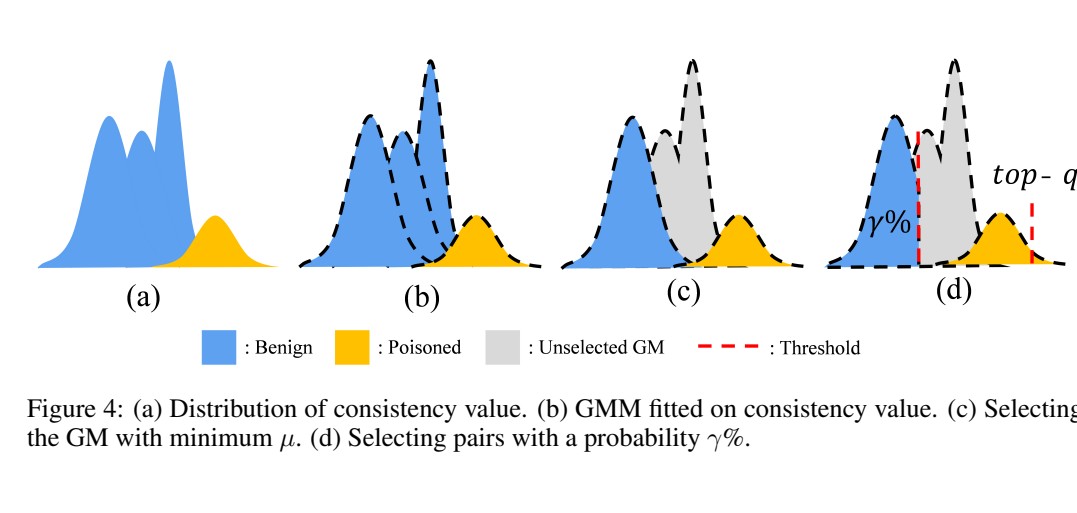

Figure 4: (a) Distribution of consistency value. (b) GMM fitted on consistency value. (c) Selecting the GM with minimum $\mu$. (d) Selecting pairs with a probability $\gamma\%$.

embedding to the textual embedding:

$$\hat{T}_i^e = F(I_i^e). \tag{1}$$

With this caption generation process, we can produce synthetic captions for each pair within the dataset $\mathcal{D}$, facilitating the calculation of the subsequent cross-modality consistency.

**Cross-modality Consistency.** We now introduce the way of calculating the cross-modality similarity $s_{cm}$, upon which the cross-modality consistency $c_{cm}$ can be derived. For a generic image-caption pair $(I, T)$, the $s_{cm}$ can be formulated as the rectified cosine similarity between the visual embedding and textual embedding, namely,

$$s_{cm}(I^e, T^e) = max\Big(\langle I^e, T^e\rangle, 0\Big), \tag{2}$$

where $\langle \cdot, \cdot \rangle$ calculates the cosine similarity. Then the original similarity $s_i^o$ of pair $(I_i, T_i)$, and the generated similarity $s_i^g$ of pair $(I_i, \hat{T}_i^e)$ can be expressed as:

$$
\begin{aligned}
s_i^o &= s_{cm}(I_i^e, T_i^e), \\
s_i^g &= s_{cm}(I_i^e, \hat{T}_i^e).
\end{aligned}
\tag{3}
$$

Since $\hat{T}_i^e$ generated based on Eq. (1) is aligned with the visual semantics $I_i$, the generated similarity $s_i^g$ has a relatively high value, regardless of poisoned pairs or benign pairs. Meanwhile, the original similarity $s_i^o$ of poisoned pairs is significantly lower than that of benign pairs, according to the unalignment between the poisoned image $\hat{I}_i$ and target caption $T_i^y$. From another perspective, benign pairs have higher consistency between $s_i^o$ and $s_i^g$, while poisoned pairs would have lower consistency. Therefore, we further define cross-modality consistency $c_i$ as the absolute difference between $s_i^o$ and $s_i^g$:

$$c_i = |s_i^g - s_i^o|. \tag{4}$$

As a result, the $c_i$ of poisoned pair is expected to be higher than that of benign pair. Next, we delve into how to detect poisoned pairs based on this consistency value $c_i$.

**Identifying the $\mathcal{D}_b$ with a GMM-based approach.** According to an experimental justification (see Figure 7a in AppendixG), the distribution of the consistency values can be appropriately modeled using a GMM consisting of multiple Gaussian distributions. We aim to select pairs from the Gaussian distribution with lower mean to avoid mistakenly choosing poisoned pairs. Specifically, the GMM being adopted is a weighted sum of $K$ Gaussian distributions:

$$p(c) = \sum_{k=1}^{K} \phi_k \mathcal{N}(c \mid \mu_k, \sigma_k), \tag{5}$$

where $\mu_k$ and $\sigma_k$ denote the mean and variance of the $k$-th Gaussian distribution. The mixture weights are defined as $\phi_k$ with the constraint that $\sum_{k=1}^{K} \phi_k = 1$. To find an optimal estimation of $\{\mu_k, \sigma_k, \phi_k\}_{k=1}^{K}$, we apply the Expectation Maximization (EM) algorithm Dempster et al. (1977).

Afterwards, we select the Gaussian distribution $\mathcal{N}(c \mid \mu_k, \sigma_k)$ with the minimum $\mu_k$. Pairs that have the probability of being from the $\mathcal{N}$ higher than a probability threshold $\gamma\%$ are categorized into the benign subset $\mathcal{D}_b$. Such a process can be illustrated in Figure 4. In addition to $\mathcal{D}_b$, we also select the top-$q$ pairs based on the consistency value $c_i$ to form the poisoned subset $\mathcal{D}_p$.

Before ending this subsection, we now briefly explain how to select appropriate values for $K$, $\gamma$, and $q$ empirically. Clearly, when $K = 1$, GMM is degraded to a single Gaussian distribution, while a larger $K$ reduces the overlapping region between the Gaussian distributions with the minimum and maximum means, thereby preventing the inclusion of poisoned pairs in $\mathcal{D}_s$. However, the coverage of each distribution could also be reduced, consequently decreasing the cardinality of $\mathcal{D}_b$. For dataset with more uniform consistency distribution, it is suggested to apply a larger $K$ to keep the Gaussian distribution with minimum mean away from that with maximum mean for achieving a high recall in $\mathcal{D}_b$. As for the non-uniform case, a smaller $K$ assists the coarse-grained detection to detect more benign pairs. We here empirically set $K = 5$, striking a good balance between these two factors. Similarly, a larger $\gamma$ leads to the inclusion of more pairs in $\mathcal{D}_b$, but at the risk of incorporating more poisoned pairs. Regarding $q$, even a relatively small value, e.g., $q = 50$ gives a satisfactory result for fine-grained detection. Also, too large $q$ would induce a drop of the detection performance.

## 2.3 FINE-GRAINED DETECTION STAGE VIA TEXTUAL EMBEDDING SIMILARITY

Given a suspicious pair $(I_u, T_u)$ from the suspicious subset $\mathcal{D}_s$, we propose to measure the average textual correlation between $T_u$ and the captions of identified poisoned data from $\mathcal{D}_p$. Such an average textual correlation would be used to identify whether the suspicious pair is poisoned or benign. Specifically, the average textual correlation $z_u$ is defined as:

$$z_u = \frac{1}{|\mathcal{D}_p|} \sum_{T_j \in \mathcal{T}_p} \langle f_T(T_u), f_T(T_j) \rangle. \tag{6}$$

Obviously, a large $z_u$ would indicate that the suspicious pair $(I_u, T_u)$ is poisoned; otherwise, it is benign. We apply an empirical threshold $\gamma_z$ on $z_u$, i.e., $z_u > \gamma_z$ means that the pair is poisoned. As expected and will be verified in Section 3.3, our fine-grained stage can detect poisoned pairs from the suspicious subset with high precision.

## 3 EXPERIMENT RESULTS OF CFBD

### 3.1 EVALUATION SETUP

**Networks.** For fair comparison between the existing solutions, we use ResNet-50 He et al. (2016) and Transformer Vaswani et al. (2017) as visual and text encoders for CLIP model. Note that it is a common practice to further fine-tune from pre-trained models Chen et al. (2020a;b); Radford et al. (2021b) as training from scratch requires a huge amount of data and computing resources. We implement the mapping network with a MLP. For fairness, the mapping network is pre-trained on COCO Chen et al. (2015); Lin et al. (2014) dataset, rather than the CC-3M.

**Datasets.** Following prior methods (Bansal et al. (2023); Yang et al. (2023a;b)) and Yang et al. (2023c), we conduct experiments on a subset of the CC-3M dataset as well as the unioned Flickr-PASCAL Young et al. (2014) and COCO datasets Chen et al. (2015).

**Backdoor attacks.** Default experiment contains 3000 poisoned pairs. We compared 9 classical and widely used backdoor attacks including (1) unimodal backdoor attacks: BadNets Gu et al. (2017), Blended Chen et al. (2017), SIG Barni et al. (2019), label-consistent Turner et al. (2019), Trojan Liu et al. (2017), WaNet Nguyen & Tran (2021) and ISSBA Li et al. (2021b); (2) backdoor attacks in SSL: the multimodal attack mmPoison Yang et al. (2023c) against MCL and BadCLIP Liang et al. (2024d). Without loss of generality, in all our experiments, we maintain the target label as "banana", a class from ImageNet-1K. For label-consistent attack, we strictly follow the setting from the previous work Bansal et al. (2023) with a poison rate of 0.05%, where the local trigger is only applied to images that their original associated caption containing "banana".

**Baselines.** We consider the widely-used backdoor defense methods including CleanCLIP Bansal et al. (2023) and RoCLIP Yang et al. (2023a). Since there are no training-time detection methods

| | Attack Types | | | | | | | | |
|---|---|---|---|---|---|---|---|---|---|
| **Methods** | **BadNets** | **Blended** | **Trojan** | **ISSBA** | **LC** | **WaNet** | **mmPoison** | **BadCLIP** | **SIG** |
| Train on Clean | 59.69 | 59.69 | 59.69 | 59.69 | 59.69 | 59.69 | 59.69 | 59.69 | 59.69 |
| No Defense | 58.69 | 59.56 | 59.74 | 58.48 | 58.28 | 59.26 | 58.62 | 58.60 | 58.87 |
| CleanCLIP Bansal et al. (2023) | 53.72 | 54.29 | 54.95 | 54.14 | **55.74** | 54.79 | 53.62 | 53.98 | 53.68 |
| RoCLIP Yang et al. (2023a) | 40.37 | 44.81 | 43.78 | 44.00 | 42.09 | 49.03 | 47.47 | 49.18 | 45.26 |
| CFBD (coarse-grained) | 31.74 | 28.93 | 29.42 | 26.90 | 32.14 | 33.15 | 30.76 | 28.46 | 29.82 |
| CFBD (coarse-to-fine-grained) | **59.21** | **57.99** | **58.99** | **59.44** | 55.11 | **59.67** | **57.11** | **57.41** | **58.57** |

Table 1: Zero-shot model performance on ImageNet1K of CFBD method along with competing backdoor defense methods against 8 backdoor attacks. The best results are **boldfaced**.

| | Attack Types | | | | | | | | |
|---|---|---|---|---|---|---|---|---|---|
| **Methods** | **BadNets** | **Blended** | **Trojan** | **ISSBA** | **LC** | **WaNet** | **mmPoison** | **BadCLIP** | **SIG** |
| No Defense | 100.0 | 100.0 | 93.11 | 50.28 | 83.58 | 99.35 | 0.16 | 98.85 | 80.38 |
| CleanCLIP Bansal et al. (2023) | 17.13 | 18.43 | 21.16 | 4.13 | 0.01 | 5.49 | 0.00 | 89.60 | 21.72 |
| RoCLIP Yang et al. (2023a) | 2.36 | 0.33 | 5.64 | 4.95 | **0.00** | 0.67 | 0.00 | 47.20 | 4.23 |
| CFBD (coarse-grained) | 0.00 | 0.00 | 0.00 | 0.00 | 36.72 | 0.00 | 0.00 | 89.27 | 0.00 |
| CFBD (coarse-to-fine-grained) | **0.00** | **0.00** | **0.00** | **0.00** | 0.84 | **0.00** | **0.00** | **0.00** | **0.00** |

Table 2: ASR results of CFBD method along with competing backdoor defense methods against 8 backdoor attacks. The best results are **boldfaced**.

in MCL scenario, we extend a backdoor detection method ABL Li et al. (2021a) from unimodal learning as a baseline.

**Metric.** Following Bansal et al. (2023), we report the zero-shot classification accuracy (CA, higher the better) on the validation set of ImageNet-1K. To verify the defense effectiveness, we evaluate the ASR (lower the better), which measures the fraction of images with the backdoor trigger that are incorrectly predicted as the target class by the model. We exclude the target class while adding triggers to images from other classes to evaluate the ASR. In addition, considering the significant imbalanced distribution between poisoned and benign pairs, we report the number of detected poisoned pairs and True Positive Rate (TPR).

## 3.2 Model Performance with CFBD

We first assess the effectiveness of our proposed method (CFBD) under various attack scenarios compared to several baseline backdoor defense methods. Two primary metrics are evaluated: (1) Zero-shot performance on ImageNet1K (see Table 1), and (2) ASR against nine types of backdoor attacks (see Table 2). The zero-shot model performance on clean data for each method is first evaluated. Row "Train on Clean" denotes a model trained on a benign dataset, whereas "No Defense" signifies a model without the application of defense methods when subjected to a poisoned dataset. The last two rows represent the models respectively trained by the benign subsets detected by the coarse-grained stage and the coarse-to-fine-grained stage (full version of CFBD). Our CFBD (coarse-to-fine-grained) method achieves superior results across most attack types, demonstrating its effectiveness. Specifically, CFBD maintains a high performance close to clean training (59.69%) with minimal degradation in most attack scenarios, achieving 59.21% in the BadNets attack, 57.99% in Blended, and 59.44% in ISSBA. Although both the coarse-grained stage and full version of CFBD can reduce ASR to 0, the CA performance of the former is much inferior compared with the latter, especially when countering strong attacks such as ISSBA and WaNet. We attribute this phenomenon to the selection of $\mathcal{D}_b$ where we apply a small $\gamma$ to select pairs from one Gaussian distribution from the fitted GMM. This design aims to preserve as many benign as possible, while maintaining a low false positive, *i.e.*, a high recall is prioritized. Consequently, only a small amount of benign pairs are identified in coarse-grained detection, which results in a degraded CA performance. This illustrates the importance of the fine-grained stage in CFBD, which can further detect more poisoned pairs from the suspicion subset, preserving the integrity of benign pairs as much as the baseline "Train on Clean". The comparable model performance, where the CA gap is less than 1%, implied the effectiveness of CFBD in detecting the poisoned pairs.

In contrast, existing defense mechanisms like CleanCLIP and RoCLIP show a more significant drop in performance, with CleanCLIP ranging between 53.72% to 55.74% and RoCLIP showing signif-

| | | Attack Types | | | | |
|---|---|---|---|---|---|---|
| | | BadNets | Blended | Trojan | ISSBA | WaNet |
| Poison Rate | Methods | TPR | TPR | TPR | TPR | TPR |
| 0.6% | ABL | 54.66% | 49.26% | 43.87% | 17.16% | 24.84% |
| | CFBD | 98.46% | 98.67% | 99.69% | 98.38% | 98.28% |
| 0.5% | ABL | 45.41% | 47.92% | 31.56% | 11.28% | 19.92% |
| | CFBD | 100.0% | 98.96% | 99.08% | 98.23% | 98.27% |
| 0.4% | ABL | 47.65% | 41.77% | 26.91% | 8.27% | 17.28% |
| | CFBD | 98.49% | 99.08% | 99.02% | 98.75% | 98.02% |
| 0.3% | ABL | 29.57% | 37.16% | 25.19% | 7.67% | 15.24% |
| | CFBD | 99.45% | 98.23% | 98.30% | 98.63% | 99.01% |

Table 3: Number of detected poisoned pairs and the TPR on the CC-3M dataset. The CFBD achieves a TPR over 98% on different attacks with different poison rates.

| | Attack Types | | | | | | | |
|---|---|---|---|---|---|---|---|---|
| Methods | BadNets | Blended | Trojan | ISSBA | WaNet | mmPoison | BadCLIP | SIG |
| CFBD | 0.9999 | 0.9999 | 0.9999 | 0.9999 | 0.9999 | 0.9999 | 0.9358 | 0.9999 |

Table 4: AUROC of CFBD against different attacks.

icantly lower scores, especially for attacks like BadNets (40.37%) and Trojan (43.78%). RoCLIP shows significantly lower scores, with about 10∼20% performance drop compared to the clean model, especially for attacks like BadNets (40.37%) and Trojan (43.78%). As also indicated in their paper Yang et al. (2023a), this is due to the replacement of original captions with the nearest neighbors in the dataset for contrastive learning. A nearest neighbor caption could have some descriptions unrelated to the target image, thereby degrading the model performance trained with RoCLIP.

When measuring the ASR, the proposed CFBD approach, particularly the coarse-to-fine-grained variant, demonstrates its superiority by successfully mitigating nearly all backdoor attacks. Our CFBD (coarse-to-fine-grained) method achieves a 0.00% ASR in seven out of the nine evaluated attack scenarios, outperforming both CleanCLIP and RoCLIP. CleanCLIP has residual vulnerabilities, particularly against the BadCLIP attack (89.60% ASR), while RoCLIP, although performing well in most cases, still exhibits higher ASR against BadCLIP (47.20%). In contrast, our method consistently reduces ASR to 0.00%, indicating a complete defense against these attack types. The coarse-grained variant of CFBD also performs robustly, though it struggles against the LC attack (36.72% ASR), highlighting the benefit of our coarse-to-fine-grained approach.

Overall, these results demonstrate that the coarse-to-fine-grained variant of CFBD provides state-of-the-art defense capabilities while maintaining high zero-shot performance, making it a promising solution for detecting diverse backdoor attacks with extremely high precision.

## 3.3 DETECTION RESULT OF CFBD

We now investigate the detection quality of the CFBD, where the detection results are tabulated in Table 3. As a poisoned data detector, CFBD can successfully detect almost all poisoned pairs across different attacks and varying poisoned rates. Specifically, in all settings, CFBD consistently detects over 98% poisoned pairs (often reaching 100% detection), while exhibiting a high recall. We also assess the effectiveness of CFBD with different poison rates ranging from 0.3% to 0.6%. The TPR and the false positive remain comparatively stable with the increase of poisoned samples. Compared with ABL, Our CFBD achieves better TPR in all settings, indicating a direct extension of defense method from unimodal learning may not lead to satisfactory results. A possible reason behind the poor detection result of ABL is: for attack with an extremely low poisoned number, the assumption in ABL that model learns the backdoor faster than the benign data is no longer valid. Additionally, we provide the AUROC results of CFBD against 8 types of attack in Table 4. Our method demonstrates the capacity to yield a satisfying detection result across all attacks. Notably, apart from the BadCLIP method, the AUROC scores with other attacks reach at 0.9999, signifying that most of the poisoned pairs are correctly detected and benign pairs are barely mis-identified.

| | Attack Types | | | | | | | |
|---|---|---|---|---|---|---|---|---|
| Size of poisoned subset | q=50 | | | | q=400 | | | |
| Fault Ratio | 20% | 40% | 60% | 80% | 20% | 40% | 60% | 80% |
| BadNets | 0.9999 | 0.9995 | 0.9893 | 0.9217 | 0.9999 | 0.9995 | 0.9993 | 0.9905 |
| Blended | 0.9999 | 0.9993 | 0.9887 | 0.9174 | 0.9999 | 0.9994 | 0.9992 | 0.9911 |
| ISSBA | 0.9999 | 0.9993 | 0.9901 | 0.9136 | 0.9999 | 0.9994 | 0.9993 | 0.9908 |
| WaNet | 0.9999 | 0.9987 | 0.9894 | 0.9184 | 0.9999 | 0.9994 | 0.9991 | 0.9909 |
| Trojan | 0.9999 | 0.9996 | 0.9885 | 0.9196 | 0.9999 | 0.9995 | 0.9989 | 0.9905 |
| SIG | 0.9999 | 0.9999 | 0.9887 | 0.9166 | 0.9999 | 0.9994 | 0.9992 | 0.9906 |

Table 5: AUROC results of CFBD when poisoned subset contains benign pairs.

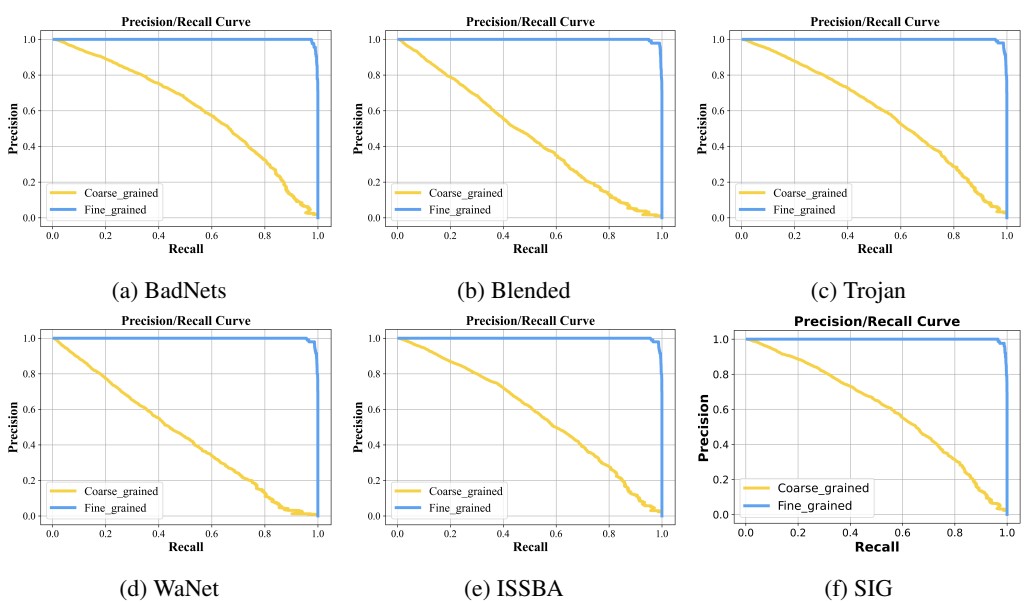

Figure 5: Precision-Recall curves of coarse- and coarse-to-fine-grained detection on six attacks. As can be seen from the curves that fine-grained detection result achieves a AUPRC approaching 1, significantly suppressing the coarse-grained results.

In this experiment, we also explore the fault tolerance of CFBD with the output poisoned subset $\mathcal{D}_p$ from the coarse-grained detection stage, in other words, how the quality of $\mathcal{D}_p$ affect the overall detection performance. Concretely, we manually construct the poisoned subset $\mathcal{D}_p$ mixed with varying ratios of benign pairs, simulating $\mathcal{D}_p$ with different extent of error. For simplicity, we evaluate this when the size of the poisoned subset $q$ is 50 or 400. The fault ratio is defined as the percentage of benign samples mixed in the poisoned subset. Specifically, we experiment on settings where the fault ratio stands at 20%, 40%, 60%, and 80%. Table. 5 presents the detection performance of CFBD. It has been observed that as more benign pairs are mixed into the poisoned subset, the AUROC displays a downward trend. For poisoned subset with 50 pairs, even when 80% of the poisoned subset are benign pairs, indicating an extreme poor detection result in the first stage, the AUROC value can still be over 0.91. When the number of poisoned pairs increase to 400, the CFBD exhibit a even better fault tolerance with AUROC value being over 0.99 under a fault ratio of 80%. These results imply that CFBD maintains a high fault tolerance with the first stage detection.

Since Precision-Recall (PR) curves give a more informative picture of an algorithm's performance than Receiver Operating Characteristic (ROC) when dealing with highly imbalanced distributions Davis & Goadrich (2006), we depict PR curves in Figure 5 for the detection results of coarse-grained and coarse-to-fine-grained CFBD. Obviously, the coarse-stage detection cannot identify all poisoned samples well, where only a high precision can be achieved when the recall is close to 0. While the coarse-to-fine-grained detection (full version of CFBD) exhibits impressive precision and recall against all attacks, with the Area Under the Precision-Recall Curve (AUPRC) approaching 1.

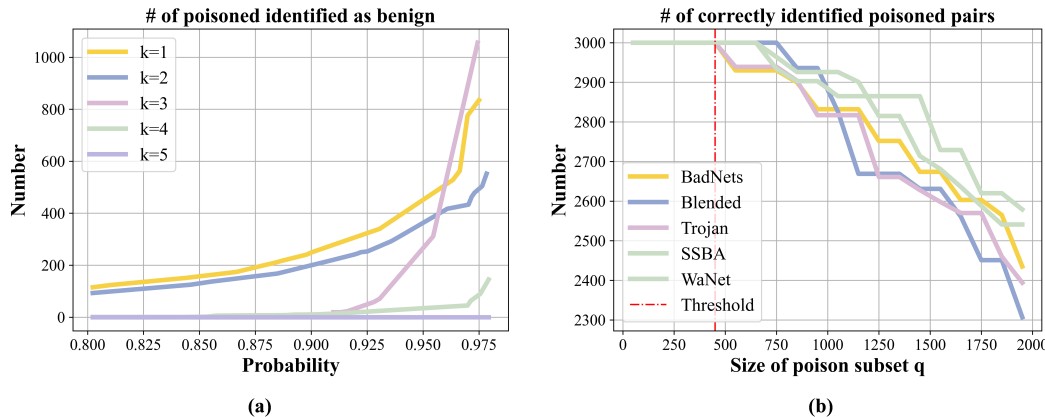

Figure 6: (a) Impact on detection results with different $K$ and $\gamma\%$ in GMM. (b) Impact on detection results using different $q$.

This indicates that all of the top-scoring pairs identified in fine-grained detection are truly poisoned. To this end, it is feasible to find a $\gamma_z$ to filter out all poisoned pairs. An example distribution of textual correlation is further provided in Appendix G.

# 4 ABLATION STUDY

Unless otherwise stated, in the ablation study below, the number of poisoned pairs is 3000.

**Different number of distribution $K$ and threshold $\gamma$ in GMM:** Figure 6 (a) reports the number of poisoned pairs being incorrectly categorized to $\mathcal{D}_b$. It is noted that the GMM is degraded to a single Gaussian distribution when $K$ is 1. In this case, over 200 poisoned pairs are erroneously included in the $\mathcal{D}_b$ when the $\gamma\%$ is higher than 87.5%. By gradually fitting GMM with more distributions, the number of misclassified poisoned pairs is remarkably reduced. When we fix the $K$, exclusively increasing the threshold $\gamma\%$ will raise the risk of preserving poisoned pairs in the $\mathcal{D}_b$ and eventually fail to prevent the injection of backdoors. These findings encourage us to set the $K$ as 5 and $\gamma\%$ as 90% to detect the benign subset for ensuring high recall.

**Different size of poisoned subset:** We now testify if increasing the $q$ in coarse-grained stage could lead to better detection performance against five different attacks. As can be seen from Figure 6 (b), our CFBD consistently succeeds to detect all poisoned pairs when $q$ is increased from 50 to 400. However, with more poisoned pairs being involved, the detection performance actually drops. We conjecture that when increasing $q$, some benign pairs would be misclassified as the poisoned subset, consequently disturb the evaluation of average textual correlation and finally deteriorates the detection results. Therefore, $q$ is empirically set as 50 in the coarse-grained stage of CFBD.

# 5 CONCLUSION

This paper proposes a coarse-to-fine detection method CFBD against backdoor attacks in MCL. We utilize the cross-modality consistency to separate the poisoned pairs and benign pairs. Extensive experiments confirm that our CFBD is capable of detecting poisoned pairs with high precision and low recall. Consequently, the CLIP model trained with benign pairs identified by CFBD maintains a extremely low ASR, meanwhile a high model performance on the test set.

CFBD makes a valuable step towards better detection of backdoor attacks in MCL. Regarding the potential threats of stronger attacks, one direction is to optimize the target caption in a way the visual semantics and textual semantics are aligned to escape the detection from CFBD. Additionally, CFBD can only be used to filter poisoned pairs in training set whereas it could not eliminate the backdoors for poisoned models. Therefore, exploring the combination of detection and unlearning strategies will be considered as our future work. *Ethical statement can be found in Appendix I.*

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
