# OpenReview forum: "CFBD: COARSE-TO-FINE DETECTION OF BACKDOOR ATTACKS IN MULTIMODAL CONTRASTIVE LEARNING"
_ICLR.cc/2025/Conference — Submitted to ICLR 2025_

### Official Review · Reviewer_3Fcp · 2024-10-28

**Soundness:** 3
**Presentation:** 2
**Contribution:** 2
**Rating:** 5
**Confidence:** 4

**Summary:**

This paper presents CFBD, a two-stage backdoor detection method for Multimodal Contrastive Learning (MCL) models, specifically those using image-caption datasets like CLIP. CFBD uses a Gaussian Mixture Model (GMM) in the coarse-grained stage to partition the dataset into subsets, and average textual correlation in the fine-grained stage to further classify suspicious pairs. Experiments show CFBD achieves impressive results across various attacks on the CC3M dataset, outperforming the baselines and maintaining high model performance on benign data.

**Strengths:**

1. **Rich Experimental Scenarios**: The authors compared 9 classical and widely used backdoor attacks, including unimodal backdoor and multimodal backdoor attacks.
2. **Detection Performance**: CFBD demonstrates superior performance in detecting most backdoor attacks.

**Weaknesses:**

1.**Lack of Novelty**: The article does not sufficiently demonstrate unique insights into multimodal backdoor attacks. It also lacks references and discussions on similar methods, such as VDC[A].
2. **Lack of comparisons**: The author does not discuss the limitations of the traititime detection method and lacks comparison with other stage methods (such as fine-tuning, pre-training) usage scenarios and purposes.
3.  **Lack of baselines**: The statement "However, the majority of existing backdoor detection methods in MCL usually produce non-satisfying detection results" lacks experimental evidence in that  the author only compared a single-modal detection method. I recommend adding more baselines for comparison.
4.  **Absence of Discussion on Training Time Limitations**: The authors do not discuss the limitations of the training time for the proposed detection method. As a detection method, detection efficiency is crucial, and the authors should provide a detailed analysis of the time requirements for their method.
5.  **Need for More Related Work**: The article should include and compare more related work, such as VDC, BadCLIP, BDetCLIP, and TA-Cleaner.
6. **Minor Issues with Figures and Tables**: I suggest the authors refine the figures and tables to enhance clarity and readability.

References:
[A]  VDC: Versatile Data Cleanser based on Visual-Linguistic Inconsistency by Multimodal Large Language Models.
[B]  BDetCLIP: Multimodal Prompting Contrastive Test-Time Backdoor Detection.
[C]  BadCLIP: Trigger-Aware Prompt Learning for Backdoor Attacks on CLIP.
[D] TA-Cleaner: A Fine-grained Text Alignment Backdoor Defense Strategy for Multimodal Contrastive Learning.
[E] Efficient Backdoor Defense in Multimodal Contrastive Learning: A Token-Level Unlearning Method for Mitigating Threats.

**Questions:**

See Weaknesses

---

### Official Review · Reviewer_F2Gc · 2024-11-03

**Soundness:** 3
**Presentation:** 3
**Contribution:** 2
**Rating:** 3
**Confidence:** 4

**Summary:**

The article discusses a two-stage backdoor detection method, CFBD, for Multimodal Contrastive Learning (MCL) that improves detection performance against backdoor attacks. By addressing the limitations of existing methods, CFBD achieves nearly 100% True Positive Rate through a coarse-to-fine approach that enhances the identification of poisoned and benign data pairs.

**Strengths:**

1. **Clear Logic and Simplicity**: The article presents its methods in a clear and logical manner, making the CFBD approach easy to understand and implement, while demonstrating its effectiveness in addressing backdoor attacks.

2. **Extensive Experiments with Strong Results**: The authors conducted extensive experiments, showcasing impressive detection performance across various attacks, which reinforces the robustness and reliability of the proposed CFBD method.

**Weaknesses:**

1. **Lack of Rigor in Motivation**: The motivation presented in the article is not sufficiently rigorous. As stated in lines 91-96, “We start by pinpointing a prevalent problem in current solutions that one-stage detection is inadequate to accurately discriminate poisoned pairs within the poisoned dataset. Specifically, these methods regularly misidentify benign pairs as poisoned pairs.” This motivation has already been proposed in previous works, such as [1], which discuss dataset partitioning and two-stage dataset handling. Therefore, the claim to “extend the mainstream one-stage detection architecture into a coarse-to-fine two-stage detection architecture” in Contribution cannot be considered a contribution of this paper.

2. **Methodological Issues**:
  * Firstly, the first stage of the proposed method relies on an additional mapping network for filtering poisoned data, which raises several concerns. This approach cannot defend against clean label attacks, and its effectiveness is contingent upon the mapping network F’s capability. If F lacks the ability to effectively distinguish between different categories, it may misclassify clean samples as poisoned ones.

   * Secondly, the suspicious subset should correspond to some hard samples that significantly contribute to the model. Since the fine-grained stage solely relies on textual similarity for detection, any clean data filtered out in the first stage could severely undermine the label accuracy in the second stage, subsequently affecting the model's performance.

* Lastly, the proposed backdoor detection method is only applicable to single-category poisoning attacks and could harm the performance of the target label class. For all-to-all poisoning scenarios, the method would classify all suspicious subsets as poisoned samples, which is harmful to the model's performance. Therefore, the decision to rely solely on textual information in the second stage, while disregarding image information, appears overly simplistic and suboptimal.

3. **Insufficient Experimental Evidence**:
   (1) The claim of achieving state-of-the-art (SOTA) results is overstated. I noticed that the related work section cites numerous new defense methods against MCL, yet the authors do not compare their method to these recent works, only contrasting it with two existing defenses (CleanCLIP and RoCLIP). Thus, claiming SOTA performance is an overstatement.
   (2) The investigation into the impact of poison rate is inadequate. The proposed method has several key hyperparameters that should be strongly correlated with the poison rate; however, the range presented in Table 3 is too narrow to demonstrate the robustness of the method. In fact, I believe that some hyperparameters related to dataset partitioning are crucial to the final outcome, and an adaptive method should be designed to allow for flexibility under varying poison rates.

4. **Suspicion of Plagiarism**:
   There appears to be significant overlap in the related work section, with numerous citations that seem unnecessary. A notable example is the citation of Liang et al.'s work on objective detection (Liang et al., 2022a). I also found that references to Liu et al. (2006) and Tang & Li (2004) are identical to those in paper [1]. Furthermore, the structure, vocabulary, terminology, citations, descriptions, and discussions in the related work section closely resemble those in [1]. Given that the introduction also cites [1], I believe this paper may have directly copied the related work from [1].

[1] Unlearning Backdoor Threats: Enhancing Backdoor Defense in Multimodal Contrastive Learning via Local Token Unlearning.

**Questions:**

Please refer to the weakness for details.

---

### Official Review · Reviewer_GFjy · 2024-11-03

**Soundness:** 3
**Presentation:** 3
**Contribution:** 2
**Rating:** 5
**Confidence:** 2

**Summary:**

Recent research highlights concerns over backdoor attacks in Multimodal Contrastive Learning (MCL) tasks, which are vital for many applications using pre-trained models. Existing detection methods often underperform due to their inability to adapt to different attack distributions and reliance on inadequate metrics like cosine similarity between images and captions. To address these issues, this work proposes a two-stage detection method called Coarse-to-Fine Backdoor Detection (CFBD), which partitions the dataset into poisoned, benign, and suspicious subsets and refines detection through average textual correlation in the second stage. CFBD achieves nearly 100% True Positive Rate (TPR) on the CC-3M dataset, significantly outperforming current methods.

**Strengths:**

Strengths.
1. The paper is clearly written and motivates the proposed approach well in a lucid manner.
2. The paper presents detailed evaluations on some datasets
3. The paper proposes a Coarse-to-Fine Backdoor Detection (CFBD) method to defend against backdoor attacks for MCL tasks.
4. The paper proposes a more effective metric based on average textual correlation, enhancing the distinction between poisoned and benign subsets.

**Weaknesses:**

Weaknesses

1. This work proposes a two-stage backdoor defense method, but the ablation of these two stages is not seen in the ablation study section.

2. The appendix section is not shown in the paper

3. In Table I, why is the zero-shot performance of the proposed method better than no defense for the backdoor attack of ISSBA?

4. This work claims, "We propose a more effective metric in the fine-grained detection stage, outperforming the widely-used image-caption similarity metric." How to demonstrate this contribution.

5. Lack of comparison with state-of-the-art backdoor detection methods in Table 5 [1][2]
[1] Xiang Z, Xiong Z, Li B. CBD: A certified backdoor detector based on local dominant probability[J]. Advances in Neural Information Processing Systems, 2024, 36.
[2] Guo J, Li Y, Chen X, et al. Scale-up: An efficient black-box input-level backdoor detection via analyzing scaled prediction consistency[J]. arXiv preprint arXiv:2302.03251, 2023.

**Questions:**

Refer to weaknesses.

---

### Official Review · Reviewer_uXn7 · 2024-11-04

**Soundness:** 2
**Presentation:** 4
**Contribution:** 2
**Rating:** 3
**Confidence:** 5

**Summary:**

This paper proposes a two-stage attack detection method for pretrained backdoored CLIP models by dividing the poisoned dataset into different subsets, namely suspicious, benign, and poisoned subsets, and improves detection quality through the average textual relevance of the poisoned subset.

**Strengths:**

1. The structure of the paper is logical, with clear main text and conclusions, making it easy for readers to understand and follow.
2. The diagrams in the paper are aesthetically pleasing and easy to read.

**Weaknesses:**

1. Lack of novelty. In lines 091-093 of the paper, the authors claim to be the first to point out the inadequacies of single-stage detection. However, this point has already been confirmed in paper [1]. Moreover, the "two-tier detection architecture" proposed by the authors is also very similar to that in paper [1], especially in the subdivision and naming of subsets: suspicious, poisoned, and benign subsets.
2. Inappropriate citations. Some references cited by the authors do not appear in the main text or related work, and it is unclear why they are listed as references. There are about 30 or more papers that are completely similar to paper [1] but are not significantly related to this paper. The authors are requested to explain this high degree of overlap in references.
3. Insufficient experimentation. Although paper [1], a highly similar relevant article, is correctly cited by the authors, no comparisons are given. Additionally, the authors should compare more recent backdoor defense methods for CLIP to demonstrate the superiority of their method.
4. Insufficient method details. The authors describe a coarse-to-fine backdoor detection framework used to distinguish samples within the poisoned dataset. However, the experimental results provided are an assessment of mitigating backdoors in CLIP (reduction in ASR). The paper lacks detailed explanation on how the detection results mitigate backdoor effects in the CLIP model, such as whether fine-tuning, retraining, or forgetting techniques were used.

Overall, although the paper cites paper [1], apart from the GMM coarse-grained detection module, it appears highly similar to paper [1] in terms of research motivation and methods. Additionally, the issue of inappropriate citations further raises questions about the originality of the paper.

Reference [1]:
Unlearning backdoor threats: Enhancing backdoor defense in multimodal contrastive learning via local token unlearning.

**Questions:**

Please refer to Weaknesses.

---

### Comment · Area_Chair_9H95 · 2024-11-22

Dear Authors and Reviewers,

The discussion phase has passed 10 days. If you want to discuss this with each other, please post your thoughts by adding official comments.

Thanks for your efforts and contributions to ICLR 2025.

Best regards,

Your Area Chair

---

### Meta-Review · Area_Chair_9H95 · 2024-12-10

**Metareview:**

This paper proposes an interesting method to study the backdoor detection. The paper has good motivation to support their study. However, as reviewers point out, the current empirical evidence cannot support the acceptance. Although a highly similar relevant article is correctly cited by the authors, no comparisons are given. Additionally, the authors should compare more recent backdoor defense methods for CLIP to demonstrate the superiority of their method. In addition, the authors describe a coarse-to-fine backdoor detection framework used to distinguish samples within the poisoned dataset. However, the experimental results provided are an assessment of mitigating backdoors in CLIP (reduction in ASR). We encourage the authors to submit a revision to other venues.

**Additional Comments On Reviewer Discussion:**

Since the authors did not provide a rebuttal and the raised concerns are consistent, there is no further discussion regarding the decision.

---

### Decision · Program_Chairs · 2025-01-22

Reject